# A Truncated Isoform of Cyclin T1 Could Contribute to the Non-Permissive HIV-1 Phenotype of U937 Promonocytic Cells

**DOI:** 10.3390/v16081176

**Published:** 2024-07-23

**Authors:** Tiziana Alberio, Mariam Shallak, Amruth Kaleem Basha Shaik, Roberto Sergio Accolla, Greta Forlani

**Affiliations:** 1Laboratory of Biochemistry and Functional Proteomics, Department of Science and High Technology, University of Insubria, 21052 Busto Arsizio, Italy; tiziana.alberio@uninsubria.it (T.A.); 2Laboratories of General Pathology and Immunology “Giovanna Tosi”, Department of Medicine and Technological Innovation, University of Insubria, 21100 Varese, Italy; mariam.shallak@uninsubria.it (M.S.); akbshaik@uninsubria.it (A.K.B.S.); roberto.accolla@uninsubria.it (R.S.A.)

**Keywords:** HIV-1, Cyclin T1, Tat-mediated HIV-1 transcription, U937 Minus, U937 Plus

## Abstract

The different susceptibility to HIV-1 infection in U937 cells—permissive (Plus) or nonpermissive (Minus)—is linked to the expression in Minus cells of interferon (IFN)-γ inducible antiviral factors such as tripartite motif-containing protein 22 (TRIM22) and class II transactivator (CIITA). CIITA interacts with Cyclin T1, a key component of the Positive-Transcription Elongation Factor b (P-TEFb) complex needed for the efficient transcription of HIV-1 upon interaction with the viral transactivator Tat. TRIM22 interacts with CIITA, recruiting it into nuclear bodies together with Cyclin T1. A 50 kDa Cyclin T1 was found only in Minus cells, alongside the canonical 80 kDa protein. The expression of this truncated form remained unaffected by proteasome inhibitors but was reduced by IFNγ treatment. Unlike the nuclear full-length protein, truncated Cyclin T1 was also present in the cytoplasm, and this subcellular localization correlated with its capacity to inhibit Tat-mediated HIV-1 transcription. The 50 kDa Cyclin T1 in Minus cells likely contributes to their non-permissive phenotype by acting as a dominant negative factor, disrupting P-TEFb complex formation and function. Its reduction upon IFNγ treatment suggests a regulatory loop by which its inhibitory role on HIV-1 replication is then exerted by the IFNγ-induced CIITA, which binds to the canonical Cyclin T1, displacing it from the P-TEFb complex.

## 1. Introduction

The mechanism of HIV-1 replication requires the exploitation of host factors, among which the P-TEFb complex, composed of Cyclin T1 and Cyclin-dependent kinase 9 (CDK9), is key for the elongation of viral primary transcript upon interaction with the viral transactivator Tat (Trans-Activator of Transcription) [1,2]. Unlike Cyclin T2A and T2B, Cyclin T1-containing P-TEFb is the sole complex that mediates HIV-1 transcription [3]. The viral transactivator Tat protein binds to Cyclin T1 and hijacks P-TEFb, recruiting it towards the Transactivation Response Element (TAR) of integrated provirus for mRNA transcription. The CDK9 subunit of P-TEFb can then hyper-phosphorylate the C- terminal domain of RNA polymerase II (RNA Pol II) along with negative elongation factors DSIF and NELF, resulting in the conversion of RNA polymerase II into a processive enzyme that initiates the entire proviral genome transcription [4,5]. Cyclin T1 is therefore a crucial co-factor in HIV-1 replication, in the absence of which the proviral transcription is impaired [6]. It has been demonstrated that Cyclin T1 acts as a key player in the reactivation of HIV-1 latency in primary CD4+ T cells [7]. CRISPR-mediated inactivation of Cyclin T1 silences HIV transcription, thus serving as an effective antiviral therapeutic target [8]. Moreover, protein kinase C (PKC) isoforms (α and β) bind to Cyclin T1 for its phosphorylation, promote the interactions between Cyclin T1 and CDK9, and enhance Cyclin T1 stability in activated and proliferating cells [9]. In contrast, Cyclin T1 is rapidly degraded in resting cells. The U937 promonocytic cell line provides a convenient model for studying HIV-1 replication, as it is characterized by the constitutive expression of CD4 and CXCR4 [10]. Based on their capacity to support CXCR4-dependent (X4) HIV-1 replication, two distinct cell clones were identified and referred to as Plus and Minus clones for permissive and non-permissive phenotypes, respectively [11,12]. However, the reason behind the restrictive HIV-1 replication in the U937 Minus cells remains partially unclear. It is believed that Interferon (IFN)γ-induced factors are potent in restricting HIV-1 during the early steps of its life cycle. Among these factors, tripartite motif-containing protein 22 (TRIM22) and class II transactivator (CIITA) were reported to be involved in repressing HIV proviral transcription. It has been previously demonstrated that TRIM22 interferes with the binding of Specific protein 1 (Sp1) to the HIV-1 LTR promoter region, thus leading to a partial inhibition of HIV transcription in a Tat- and NF-kB independent manner [13]. In addition, CIITA competes with Tat for binding to the Cyclin T1 subunit of P-TEFb, suppressing the viral replication [14,15]. Interestingly, both TRIM22 and CIITA were found to be expressed in HIV-1 non-permissive Minus cells but not in HIV-1 permissive Plus cells. Moreover, CIITA-forced expression in Plus cells inhibited Tat-dependent HIV-1 replication independent of TRIM22 [16]. Interestingly, we have recently shown the existence of a physical colocalization between TRIM22 and CIITA with promyelocytic leukemia (PML) protein in newly identified nuclear bodies [17]. Endogenous Cyclin T1 is also recruited in CIITA-TRIM22-containing nuclear bodies. All these crucial factors physically interact and form molecular complexes acting as a negative regulator of proviral transcription. 

P-TEFb is differentially expressed in CD4+ T lymphocytes, monocytes, and macrophages, depending on their activation and differentiation status. PMA treatment, a stimulus inducing the differentiation of U937 promonocytic cells into macrophage-like cells, results in a reduction in CIITA expression and in increased levels of HIV-1 replication and Cyclin T1 expression. It is conceivable that the inhibition of Tat activity in Minus cells may be influenced not only by the sequestration of Cyclin T1 by CIITA but also by the modulation of Cyclin T1 expression itself. In this study, we identified a distinct pattern of Cyclin T1 expression in Minus cells compared to that of Plus cells. Immunoprecipitation of Cyclin T1 protein in Minus cells followed by mass spectrometry analysis revealed an enrichment of a 50 kDa Cyclin T1 protein form over the canonical 80 kDa protein. This pattern of Cyclin T1 expression was neither affected by the inhibition of the proteosomal degradation nor the result of a splicing event, suggesting that Cyclin T1 variants might result from a cleavage by a specific cellular protease. This Cyclin T1 protein form, unlike the nuclear full-length protein, is also localized in the cytoplasmic compartment. The cytoplasmic distribution of 50 kDa Cyclin T1 protein form correlates with its capacity to inhibit Tat-mediated HIV-1 LTR transcription, suggesting that it could act as dominant negative factor limiting the formation and function of the PTEF-b complex.

## 2. Materials and Methods

### 2.1. Plasmids

pHA-Cyclin T1 vector was a gift from M. B. Peterlin (UCSF). Plasmid encoding for the HA-tagged truncated form of Cyclin T1 (first 481 amino acids) was generated by Vector Builder (pHA-CyclinT1-50 kDa). pTat vector was previously described [16].

### 2.2. Cells

U937 isogenic Minus and Plus, U937 Plus-fCIITA (1F6 clone), and Jurkat T cells were previously described [15,16]. U937 Minus, U937 Plus, and Jurkat cells were grown in RPMI-1640 medium supplemented with 10% heat-inactivated fetal calf serum (FCS) and 5 mM l-glutamine (complete medium). U937 Plus-CIITA cells were grown in complete RPMI-1640 under 1 mg/mL of G418 (Sigma Chemical Corp., St. Louis, MO, USA) selection. Human embryonic kidney 293T cells (kindly provided by Prof. B.M. Peterlin, UCSF, San Francisco, CA, USA) were cultured in Dulbecco’s modified Eagle medium containing 5 mM l-glutamine and supplemented with 10% FCS. 

### 2.3. Immunoprecipitation and Western Blotting

Endogenous Cyclin T1 protein was precipitated from 5 × 10^6^ Minus and Plus cells. Cells were lysed on ice for 45 min with lysis buffer (1% NP-40, 10 mM Tris–HCl pH 7.4, 150 mM NaCl, 2 mM EDTA) supplemented with 0.1% protease inhibitor cocktail (Sigma). After pre-clearing with protein A-Sepharose beads, whole-cell extracts were incubated in ice with rabbit-polyclonal anti-CCNT1 antibody (Sigma-Aldrich, HPA004892) for 1 h and then immunoprecipitated with protein A-Sepharose beads overnight at 4 °C. The precipitated proteins were resolved on 9% SDS-PAGE and analyzed by Western blotting with the anti-CCNT1 antibody or with the anti-CDK9 antibody (Cell Signaling, C12F7, Danvers, MA, USA), followed by horseradish peroxidase (HRP)-conjugated goat anti-rabbit IgG (Invitrogen). Blots were developed by chemiluminescence assay (Thermofisher, Waltham, MA, USA). We analyzed 8% of the precleared cell lysate for the expression of endogenous Cyclin T1 and CDK9 by Western blotting with anti-CCNT antibody or with anti-CDK9 antibody (input). Endogenous α-tubulin (housekeeping) was detected by using anti-α tubulin monoclonal antibody (1:1000) (T5168; Sigma) followed by (HRP)-conjugated anti-mouse secondary antibodies. 

### 2.4. RT-PCR

Total RNA was extracted from cell pellets using TRIzol reagent (Thermo Fisher Scientific, Waltham, MA, USA). cDNA was synthesized from 1 μg total RNA using iScript cDNA Synthesis Kit (Bio-Rad, Hercules, CA, USA). One μg of cDNA was amplified by PCR using 2×PCR Bio Taq Mix (PCR Biosystems, London, UK) according to the manufacturer’s protocol and used for subsequent RT-PCR, using the primers listed in Table 1. PCR products were analyzed by agarose gel electrophoresis with ethidium bromide under UV light. The RT-PCR experiments were performed in replicate.

The primers used to amplify the CCNT1 (Cyclin T1) gene were previously described and are listed in Table 1 [18]. In detail, the 5′ half of Cyclin T1 was amplified by RT-PCR using the T1-FW and T1-9RW primer pairs. The expected RT-PCR product for the CCNT1 full-length transcript was 1274 bp. The 3′ half of C T1 was amplified by RT-PCR using the T1-9FW and T1-RW primer pairs. The expected RT-PCR product for the CCNT1 full-length transcript was 1468 bp. The region from exon 6 to the beginning of exon 9 was amplified by RT-PCR using the T1-6FW and T1-9RW primer pairs. The expected RT-PCR product for the CCNT1 full-length transcript was 600 bp. The RT-PCR products were separated by agarose gel electrophoresis and visualized by ethidium bromide. 

### 2.5. Mass Spectrometry

Cyclin T1 immunoprecipitate from Minus cells was separated by SDS-PAGE on a 9% polyacrylamide gel, then stained with Coomassie Brilliant Blue. The excised 50 kDa band was washed (25 mM NH_4_HCO_3_ in 50% CH_3_CN), dehydrated (100% CH_3_CN), and submitted to reduction (10 mM DTT in 100 mM NH_4_HCO_3_ for 45 min at 56 °C) and alkylation (55 mM IAA in 100 mM NH_4_HCO_3_ for 30 min at RT in the dark). The gel band was then washed (25 mM NH_4_HCO_3_ in 50% CH_3_CN), dehydrated (100% CH_3_CN), and incubated for protein digestion at 37 °C overnight with 10 ng/μL trypsin in 100 mM NH_4_HCO_3_ (Modified Porcine Trypsin, mass-spectrometry grade, Promega, Madison, WI, USA). Upon digestion, peptides were extracted from gel by sequential washes with 0.1% trifluoroacetic acid (TFA)/60% CH_3_CN, collected in a fresh tube, and vacuum dried. 

The peptide mixtures were separated using an LTQ XL-Orbitrap ETD equipped with a HPLC NanoEasy-PROXEON (Thermo Fisher Scientific). Protein identification was performed by searching the National Center for Biotechnology Information non-redundant database using the Mascot search engine (http://www.matrixscience.com (accessed on 8 March 2018)). Input search parameters were set as follows. Enzyme: trypsin; fixed modifications: carbamidomethyl (C); variable modifications: oxidation (M), Gln->pyro-Glu (N-term Q), pyro-carbamidomethyl (N-term C); peptide mass tolerance: ±10 ppm; fragment mass tolerance: ±0.6 Da; maximum number of missed cleavages: 1. Uniprot release 2018_01 was used as the reference human protein sequence database. The proteins identified were ranked based on the emPAI score, and the protein showing the highest emPAI score and protein sequence coverage was retained as the protein spot identifier. Keratins and trypsin were excluded from the protein lists. 

### 2.6. Treatments

U937 Minus and U937 Plus cells were treated for 60 h with IFNγ (1000 U/mL) or its vehicle, then lysed and analyzed by Western blotting to assess Cyclin T1 expression, as described above. U937 Minus and Plus cells were incubated for 6 h with MG132 30 μM or its vehicle (DMSO), then lysed and analyzed by Western blotting to assess Cyclin T1 expression, as described above.

### 2.7. Immunofluorescence and Confocal Microscopy

Human embryonic kidney 293T cells seeded on glass coverslips were transfected with the expression vector for either full-length Cyclin T1 or truncated Cyclin T1 50kDa by FugeneHD (Promega). After 24 h, the cells were fixed with cold methanol, washed three times with -PBS, and blocked for 1 h with -PBS containing 0.5% bovine serum albumin (Sigma). The cells were stained overnight with mouse monoclonal anti-HA (Sigma-Aldrich, H7 clone, diluted 1:400 in -PBS buffer containing 0.1% bovine serum albumin (BSA)) at 4 °C. The slides were then washed three times with -PBS and incubated for 2 h at RT in the dark with the goat anti-mouse coupled to Alexa Fluor 546 (1:400) secondary antibody. The nuclei were then stained by incubating the cells with DRAQ5 Fluorescent Probe (1:1000) (Thermo Fisher Scientific) for 30 min at RT in the dark. After washing, the coverslips were mounted on glass slides (Thermo Fisher Scientific) with the Fluor Save reagent (Calbiochem, San Diego, CA, USA) and examined by a confocal laser scanning microscope (Leica TCS SP5, 63× original magnification, numerical aperture 1.25). Images were acquired and analyzed by LAS AF Lite Image (Leica Microsystems, Wetzlar, Germany). 

### 2.8. Transient Transfection and HIV-1-LTR Luciferase Assay

A total of 293T cells were seeded in 60 mm-diameter plates and transfected with 0.6 μg of pHIV-1 LTRLuc, 10 ng of pTat, and increasing amounts of pHA-Cyclin T1-50kDa plasmid DNA, as indicated in the corresponding figure legend, using FugeneHD (Promega). All the transfections were carried out in the presence of 0.15 μg of phRL-CMV expressing the *Renilla* luciferase. An empty pcDNA3 vector was used as a stuffer DNA to maintain constant the total amount of transfected DNA. Cell extracts were prepared 24 h post-transfection and assayed for luciferase activity by using a dual luciferase reporter assay system (Promega) according to the manufacturer’s instructions. Mean luciferase values, normalized to the *Renilla* values, of at least three independent experiments performed in duplicate are expressed as relative luciferase activity. Error bars represent the standard deviation (SD). Statistically significant values (*p* < 0.05) were assessed with a one-tailed Student *t* test. Cell lysates were analyzed for the expression of truncated Cyclin T1 proteins by SDS-PAGE and Western blotting with the anti-CCNT1 (Sigma-Aldrich) rabbit polyclonal antibody.

### 2.9. Statistical Analysis

The statistical analysis was performed using the GraphPad Prism software 10.0 (GraphPad Software, http://www.graphpad.com, accessed on 1 June 2024). Comparison between two groups was performed by using the unpaired *t* test. A *p* value < 0.05 was considered significant. 

## 3. Results

### 3.1. Cyclin T1 Is Mainly Expressed as a 50 kDa Protein in U937 Minus Cells

We previously found that the different susceptibility to HIV-1 infection in the two isogenic promonocytic U937 cell Minus and Plus clones was associated with the expression of both CIITA and TRIM22 restriction factors in Minus cells but not in HIV-1 permissive Plus cells [16,17]. In search for additional factors limiting the productive HIV-1 infection in U937 Minus cells, we evaluated the expression of the Cyclin T1 component of P-TEFb complex by Western blot. In addition to the canonical 80 kDa band corresponding to full-length Cyclin T1, Minus cells exhibited at least two additional bands of 50 kDa and 46 kDa (Figure 1A, lane 2). In contrast, U937 Plus cells displayed only the full-length protein, consistent with what we observed in Jurkat T cells (Figure 1A, lanes 1 and 3, respectively). Based on our previous observation that the expression of CIITA in Plus cells (Plus-CIITA) significantly reduced the Tat transactivating function, mimicking a Minus-like phenotype, we investigated whether the ectopic expression of CIITA in Plus cells could also modify the expression pattern of Cyclin T1 protein. As shown in Figure 1B, in Plus-CIITA cells we found only the full-length Cyclin T1 (Figure 1B, lane 3).

### 3.2. Mass Spectrometry Analysis of 50 kDa Protein Form Shows It Is Cyclin T1

In order to verify whether the most abundant band found in Minus cells (50 kDa) corresponded to Cyclin T1 protein, mass spectrometry analysis was performed (Figure 2A). The band corresponding to 50 kDa was excised from an SDS-PAGE and identified by LC-MS/MS (Appendix A). 

Mass spectrometry analysis confirmed that the 50 kDa band corresponds to Cyclin T1 (O60563), whose identification showed the highest score and emPAI (50 significant matches). All the peptides witnessed the presence of the N-terminal of the protein till the 481 residue (Figure 2B), thus containing the cyclin box (aa 1-254), which includes the CDK9-interaction domain. To further verify whether the 50 kDa CyclinT1 retained the ability to interact with CDK9, we immunoprecipitated endogenous Cyclin T1 from Minus and Plus cells and assessed the presence of CDK9 by Western blot (Figure 2C). The results indicate the presence of CDK9 in Cyclin T1 immunoprecipitate from both Minus and Plus cells, suggesting that the 50 kDa form of Cyclin T1 could indeed interact with CDK9, a crucial component of the P-TEFb complex [19,20], thereby confirming the mass spectrometry findings (Figure 2C). Moreover, the other lower-molecular-weight bands (46 kDa and 32 kDa) immunoprecipitated with the Cyclin T1-specific antibody in Minus cells were also analyzed. Interestingly, all bands were enriched with peptides corresponding to the N-terminal region of Cyclin T1, containing the binding site for CDK9. LC-MS/MS data are reported in Appendix A.

### 3.3. The 50 kDa Cyclin T1 Protein Form Does Not Result from Alternative Splicing Nor from Proteasome-Dependent Degradation

To assess whether the 50 kDa Cyclin T1 protein form was the result of a splicing event, RT-PCR was performed to amplify the CCNT1 open reading frame (ORF) using RNA extracted from U937 Minus and Plus cells. Primers targeting different regions of the ORF were utilized (Figure 3A). The T1-FW and T1-9RW primer set was used to amplify the 5′ half of the CCNT1 ORF, including exons 1 to 8 and part of exon 9. One band with the size estimated by the ORF sequence (1274 bp) was detected in all the samples after agarose gel electrophoresis (Figure 3B, left panel), indicating that the mRNA of Cyclin T1 in Minus cells was not differentially processed compared to Plus cells. Consistently, the T1-6FW and T1-9RW primer set yielded one PCR amplicon of 600 bp in all samples, consistent with the size of the corresponding ORF (Figure 3B, middle panel). Finally, the T1-9FW and T1-RW primer set was used to amplify the 3′ half of CCNT1 ORF containing exon 9. One amplicon with the size estimated by the ORF sequence (1468 bp) was found in both Minus and Plus c-DNA samples, indicating that all of exon 9 was transcribed (Figure 3B, right panel). These results demonstrate that the 50 kDa Cyclin T1 protein form did not result from a splicing event at the RNA level and suggest that this protein form could be generated by protease cleavage. Because the ubiquitin–proteasome system is one the major degradative protein pathways in mammalian cells, we next investigated whether the 50 kDa Cyclin T1 protein form in Minus cells resulted from proteasomal degradation. Treatment of cells with the proteasome inhibitor MG132 did not affect the amount of truncated Cyclin T1, suggesting that in U937 Minus cells Cyclin T1 was not degraded in a proteasome-dependent manner (Figure 4). 

These findings suggest that a specific protease active in U937 Minus cells might be responsible for the generation of the truncated protein form of Cyclin T1. 

### 3.4. IFNγ Treatment Reduces the Amount of 50 kDa Cyclin T1 Protein Form in U937 Minus Cells

Experiments were then performed to investigate whether the 50kDa Cyclin T1 protein form in Minus cells could be differentially expressed upon treatment with IFNγ, known to modulate the expression of both TRIM22 and CIITA [13,21,22]. Remarkably, treatment of Minus cells with the cytokine resulted in a significant increase in the full-length protein and a simultaneous decrease in the 50kDa protein form (Figure 5, lanes 1 and 2). In Plus cells, the canonical 80kDa Cyclin T1 was consistently detected regardless of treatment. It should be noted that these cells lack the IFNγ receptor and thus do not respond to IFNγ stimulation (Figure 5, lanes 3 and 4) [23]. 

### 3.5. The 50 kDa Cyclin T1 Protein Form Is Localized in Both the Nucleus and the Cytoplasm

To further characterize the newly identified 50 kDa Cyclin T1 protein form, we generated an expression vector coding for the first 481 aa of Cyclin T1, tagged with hemagglutinin (pCyT1HA-50 kDa). The expression of the Cyclin T1 HA-50 kDa protein was confirmed by Western blot analyses using anti-Cyclin T1 antibody. As shown in Figure 6A in 293T cells transfected with pCyT1HA-50 kDa, a protein of about 50 kDa was observed by using an anti-Cyclin T1 antibody, in addition to the full-length Cyclin T1 expressed in both Minus and Plus cells (Figure 6A, lanes 1, 2, and 3). The size of the protein corresponding to the ectopically expressed 50 kDa Cyclin T1 protein form in 293T cells was similar to that of 50 kDa Cyclin T1 protein form expressed in Minus cells (Figure 6A, compare lane 1 with lane 2). We next analyzed the intracellular distribution of HA-tagged 50 kDa Cyclin T1 in 293T cells and compared it with that of HA-tagged full-length Cyclin T1. 

Consistent with previous reports, the full-length protein was distributed to the nuclear matrix-attached speckle-domains, which are known to be the sites for the RNA Pol II-mediated transcription and the co-transcriptional pre-mRNA processing in which the P-TEFb complex is involved [18,24] (Figure 6B, upper panels). Conversely, the 50 kDa Cyclin T1 protein form was detected both in the nucleus and in the cytoplasm. Notably, its nuclear distribution was diffuse and showed very limited accumulation in discrete speckle compartments (Figure 6B, lower panels).

### 3.6. Inhibition of Tat-Mediated HIV-1 LTR Transactivation by the 50 kDa CyclinT1 Protein Form

We previously demonstrated the existence of efficient Tat-dependent transactivation of HIV-1 transcription in U937 Plus cells, while it was significantly diminished in U937 Minus cells [16]. To further investigate whether the distinct subcellular localization of the 50 kDa Cyclin T1 isoform might impact Tat-mediated HIV-1 gene transcription, we assessed Tat-dependent luciferase activity in 293T cells transiently transfected with the pHIV-1LTR-Luc reporter gene, the Tat-expressing vector, and increasing amounts of pCyT1HA-50 kDa DNA plasmid. Consistent with our hypothesis, the HIV-1 Tat/LTR-driven luciferase activity decreased in the presence of the 50 kDa Cyclin T1 protein in a dose-dependent manner (Figure 7). These results further corroborate the idea that the newly described 50 kDa Cyclin protein form acts as a potential inhibitor of HIV-1 replication. 

## 4. Discussion

U937 Plus and Minus cells represent an interesting model to study the mechanism limiting productive HIV-1 replication in the monocytic/macrophage cell lineage [25]. We previously demonstrated that CIITA, the master regulator of MHC-II gene transcription [21,26,27], is expressed in U937 Minus but not in U937 Plus cells and acts as a negative transcriptional regulator of HIV expression by targeting Tat, thus contributing to the HIV-1 non-permissive phenotype of promonocytic U937 Minus cells. Indeed, the exogenous expression of CIITA in U937 Plus cells suppresses HIV-1 transcription [16]. Remarkably, we previously showed that CIITA inhibits HIV-1 replication in CD4+ human T cell lines by competing with Tat for binding to Cyclin T1 of the P-TEFb complex [15]. 

We have described here a new truncated protein form of Cyclin T1 of about 50 kDa that we believe bears functional importance. This Cyclin T1 protein form is expressed in HIV-1 non-permissive cells and behaves as a *bona fide* inhibitor of the Tat-mediated HIV-1 transactivation. Using a polyclonal antibody against the endogenous Cyclin T1, two bands of about 50 and 46 kDa were identified in Minus cells, in addition to the 80 kDa band corresponding to the full-length Cyclin T1. Importantly, in Minus cells the 50 kDa band was much more abundant than the amount corresponding to the full-length Cyclin T1. 

Previous investigations have shown the existence of Cyclin T1 protein forms distinct from the canonical 80 kDa Cyclin T1. For example, a spliced variant of Cyclin T1 that negatively regulates the transcription elongation of HIV-1 genes was identified [18,28]. Gao and colleagues suggested that the serine–arginine-rich protein ASF/SF2 is a regulatory factor of the alternative splicing of the Cyclin T1 gene, promotes the production of Cyclin T1b variant versus full-length Cyclin T1, and regulates the expression of Cyclin T1-depedent genes at the transcription level [29]. The Cyclin T1 spliced variant has an approximate 20 kDa molecular weight; is composed of 180 amino acids, including the N-terminal cyclin box of Cyclin T1 and four additional amino acids; and retains the ability to interact with CDK9, but not with Tat. Consequentially, the complex of the spliced Cyclin T1 and CDK9 cannot be utilized by TAT/TAR for the activation of transcription elongation, rendering HIV-1 replication less efficient. Indeed, the spliced Cyclin T1 was shown to act as a dominant–negative inhibitor of the full-length Cyclin T1 by competing for CDK9.

The 50 kDa protein form described in the present investigation and observed in Minus cells did not result from alternative splicing events of the mRNA. Moreover, the 50 kDa Cyclin T1 did not result from the general proteosomal activity in Minus cells, suggesting that it is generated by proteolytic digestion with a specific protease active in Minus cells but not in Plus cells, whose nature remains to be identified. Studies suggest that nuclear proteases are endowed of a high degree of specificity for nuclear proteins. For example, nuclear cathepsins have been found to target and cleave particular nuclear substrates [30]. Numerous proteases are found in both the nucleus and the cytoplasm, though the reasons for the nuclear translocation of certain proteases remain unclear. Detailed protein sequence analysis has uncovered that some proteases have a nuclear translocation signal (NLS). Bioinformatic analysis of matrix metalloproteinases (MMPs) has demonstrated that all members of this group possess one or more NLSs [31]. Worthy of note is that the Cyclin T1 sequence has many putative cleavage sites for both cathepsin and MMPs. Moreover, in the Intact database a direct interaction between Cyclin T1 and Cathepsin E is reported [32]. 

Identification by mass spectrometry showed that the 50 kDa Cyclin T1-truncated protein included the first 481 amino acids of the Cyclin T1, containing the cyclin box and the CDK9 binding domain [33,34]. Differently from the spliced Cyclin T1 variant described above, the newly identified protein form also contained the first 272 amino acids of Cyclin T1 including the Tat–TAR recognition motif (TRM) (amino acids 250-263) and the N-terminal region (amino acids 26-56) crucial for forming the Tat–Cyclin T1–TAR ternary complex [35]. 

Although the new Cyclin T1 variant includes regions responsible for Tat-mediated activation of HIV-1 transcription, subcellular localization experiments conducted in 293T cells transfected with the 50 kDa Cyclin T1 showed that this variant is not exclusively expressed in the nucleus, like full-length Cyclin T1, but is also distributed in the cytoplasmic compartment. This suggests that the cytoplasmic pool of Cyclin T1 may not be available for interaction with CDK9 and Tat, thus limiting HIV-1 transcription. Furthermore, the nuclear portion did not accumulate in the discrete speckle compartments as full-length Cyclin T1, again suggesting a distinct role with respect to the canonical Cyclin T1. The lower molecular weight of the newly generated CyclinT1HA, compared to the truncated CyclinT1 expressed in Minus cells, is likely due to our consideration of only the first 481 amino acids included in the tryptic peptides. However, we hypothesize that the 50 kDa protein form of Cyclin T1 sequesters CDK9 in a non-functional complex. Consistent with this hypothesis, the 50 kDa Cyclin T1 protein form was found to have an opposite effect with respect to full-length Cyclin T1 in the regulation of HIV-1 transcription, since it inhibited in a dose-dependent manner the Tat-mediated LTR transactivation measured by a luciferase reporter assay. These results strongly suggest that the expression of the 50 kDa Cyclin T1 protein form represents a mechanism limiting HIV-1 transcription, because the pool of canonical nuclear Cyclin T1 available for interaction with CDK9 and Tat not only is significantly reduced but also can be competed with the action of the 50 kDa protein form. Within this frame, it has been shown that deprivation or downregulation of Cyclin T1 could lead to a downregulation of Tat-mediated HIV-1 transactivation [6,36]. 

Upon stimulation of Minus cells with IFNγ, a drastic reduction in 50 kDa Cyclin T1 and an increase in full-length Cyclin T1 was observed. Therefore, one should expect that this would favor Tat-mediated HIV-1 transactivation in Minus cells. However, we have shown that this is not the case and that, actually, IFNγ treatment results in the inhibition of HIV replication [37,38,39]. This is due particularly to the strong upregulation of CIITA, which, as previously mentioned, is a strong inhibitor of the Tat-dependent HIV-1 transcription because it competes with Tat for binding to canonical 80 kDa Cyclin T1 [15].

Thus, it is possible that the novel 50 kDa Cyclin T1 protein form and CIITA reported herein mediate similar complementary mechanisms to limit HIV-1 replication. Both reduce the availability of the canonical Cyclin T1 to be used in the P-TEFb complex for the Tat-mediated transcript elongation of the HIV-1 genome in the resting and in the IFNγ-stimulated state, respectively.

Taken together, these findings underscore the role of Cyclin T1 as a key regulator of Tat-mediated HIV-1 transcription and elucidate another mechanism utilized by U937 Minus cells to maintain their poorly permissive phenotype to HIV-1 infection. 

## Figures and Tables

**Figure 1 viruses-16-01176-f001:**
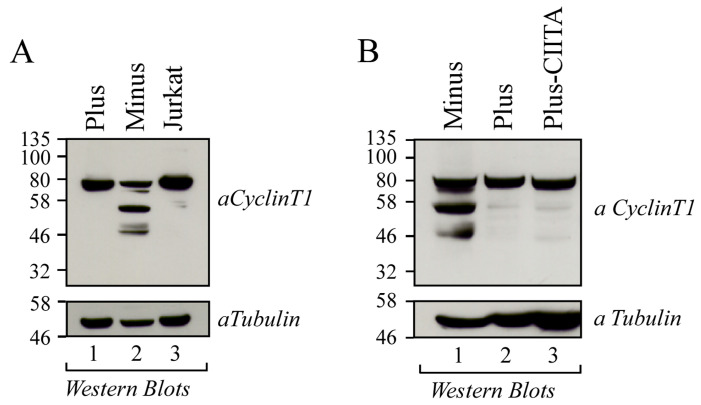
A 50 kDa Cyclin T1 protein form is mainly expressed in U937 Minus cells. (**A**) Cell lysates obtained from U937 Plus cells (6 × 10^6^ cells) (lane 1), U937 Minus cells (6 × 10^6^ cells) (lane 2), and Jurkat T cells (6 × 10^6^ cells) (lane 3) were analyzed for the presence of Cyclin T1 by Western blotting using anti-Cyclin T1 rabbit polyclonal antibody. (**B**) Cell lysates obtained from U937 Minus cells (6 × 10^6^ cells) (lane 1), U937 Plus cells (6 × 10^6^ cells) (lane 2), and U937 Plus-CIITA cells (6 × 10^6^ cells) (lane 3) were analyzed for the presence of Cyclin T1 by Western blotting. The expression of endogenous α-tubulin, used as an internal control of sample loading, was evaluated by anti-α tubulin Western blotting.

**Figure 2 viruses-16-01176-f002:**
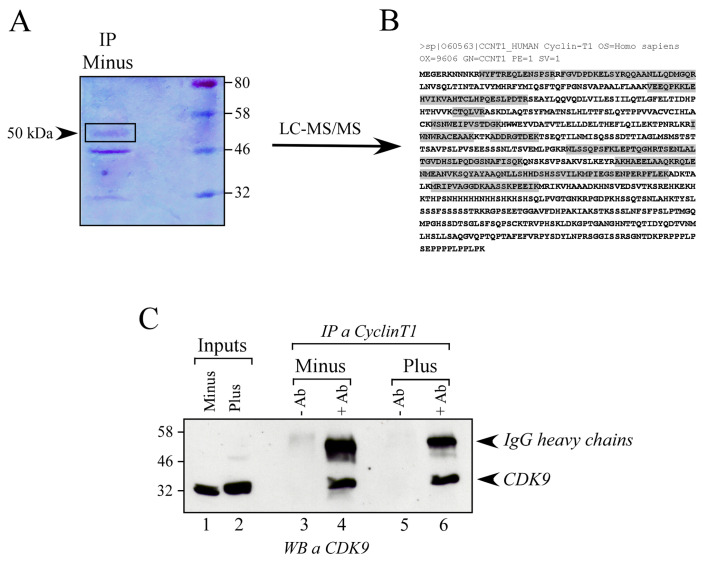
Identification of the 50 kDa band by mass spectrometry. (**A**) SDS-PAGE of the IP of Cyclin T1 in U937 Minus cells. (**B**) Human Cyclin T1 sequence (O60563). Peptides identified by LC-MS/MS are evidenced. (**C**) IP of Cyclin T1 in Plus and Minus cells. The presence of CDK9 has been detected with rabbit monoclonal anti-CDK9 antibody.

**Figure 3 viruses-16-01176-f003:**
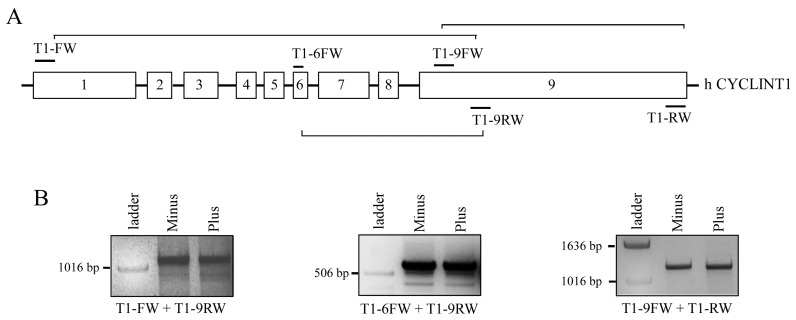
Detection of the CCNT1 transcript in U937 Plus and Minus cells by RT-PCR. (**A**) Gene structure of CCNT1. The exons and introns are shown as boxes and lines, respectively. The number of each exon is indicated inside the box. The binding sites for the primers used in this study are shown as lines on top of the exons. (**B**) RT-PCR analysis using the T1FW/T1-9RW, T1-6FW/9-RW, and T1-9FW/T1-RW primer pairs. The bands represent the full length of CCNT1.

**Figure 4 viruses-16-01176-f004:**
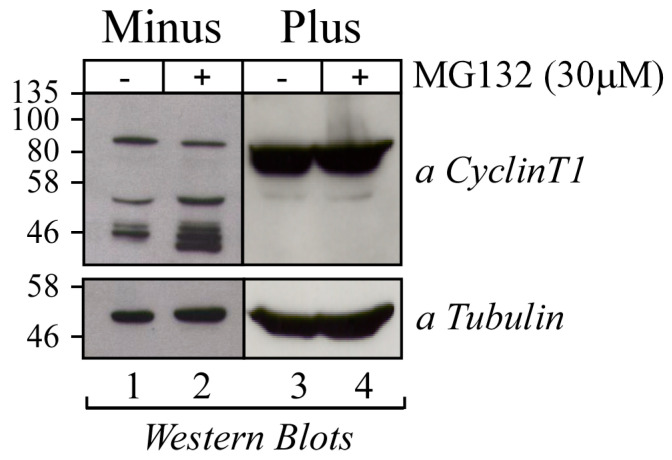
The 50 kDa Cyclin T1 did not result from the activation of proteasomal degradation. U937 Minus cells were incubated with proteasome inhibitor MG132 (30 mM) (+) or with vehicle (-) for 6 h and analyzed by Western blotting for the presence of Cyclin T1 with the anti-Cyclin T1 rabbit polyclonal antibody. The expression of endogenous α-tubulin, used as an internal control of sample loading, was evaluated by anti-α tubulin Western blotting.

**Figure 5 viruses-16-01176-f005:**
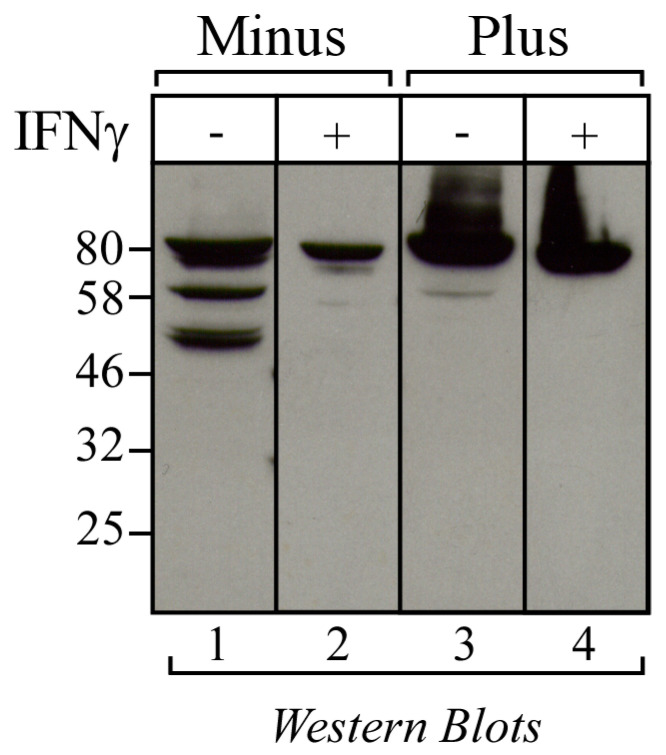
IFNγ treatment reduces the level of 50 kDa Cyclin T1 in U937 Minus cells. U937 Minus (lanes 1, 2) and Plus (lanes 3, 4) were incubated with IFNγ (1000 U/mL) (+) or with vehicle (-) for 60 h and analyzed by Western blotting with the anti-Cyclin T1 rabbit polyclonal antibody.

**Figure 6 viruses-16-01176-f006:**
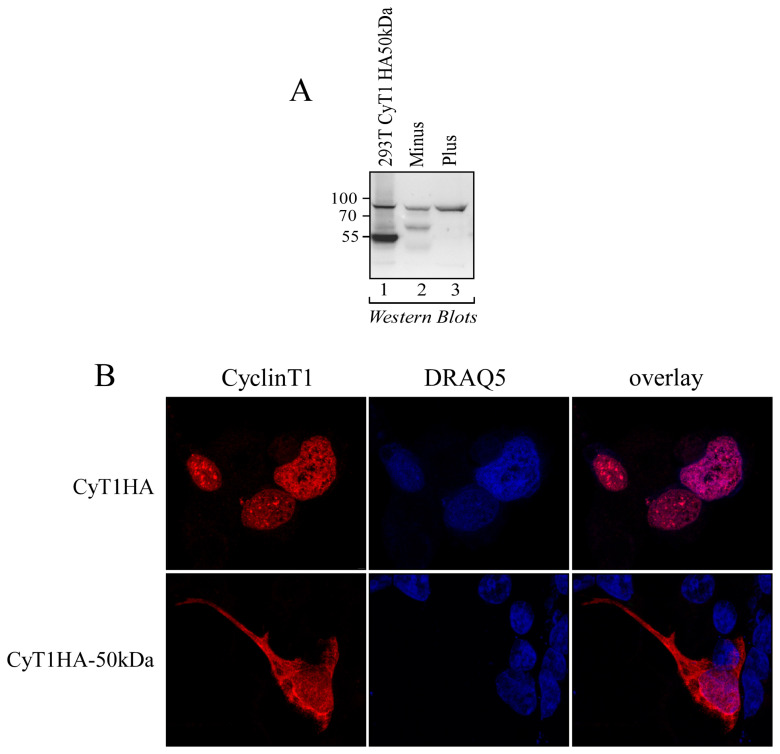
The exogenous expression of plasmid coding for HA-tagged 50 kDa Cyclin T1 resulted in a 50 kDa protein expressed both in the nucleus and in the cytoplasm. (**A**) Cellular lysates obtained from 293T cells transiently transfected with HA-tagged 50 kDa Cyclin T1 (293T Cyclin T1 HA 50 kDa, lane 1), U937 Minus (lane 2), and U937 Plus (lane 3) cells were analyzed by Western blotting for the presence of Cyclin T1 with the anti-Cyclin T1 polyclonal antibody. (**B**) The 293T cells were transfected with either HA-tagged 50 kDa Cyclin T1 (CyT1HA-50kDa) or full-length, HA-tagged Cyclin T1 (CyT1HA). Cells were fixed and stained with anti-HA mouse monoclonal antibody followed by goat anti-mouse AlexaFluor 546-conjugated antibody to detect HA-tagged Cyclin T1 (Cyclin T1, left panels). Nuclei were stained by using DRAQ5 reagent (DRAQ5, middle panels). Overlay images (right panels) were obtained by merging the images resulting from the HA staining with those obtained with DRAQ5 staining. Stained cells were then analyzed by confocal microscopy, as described in the Section 2.

**Figure 7 viruses-16-01176-f007:**
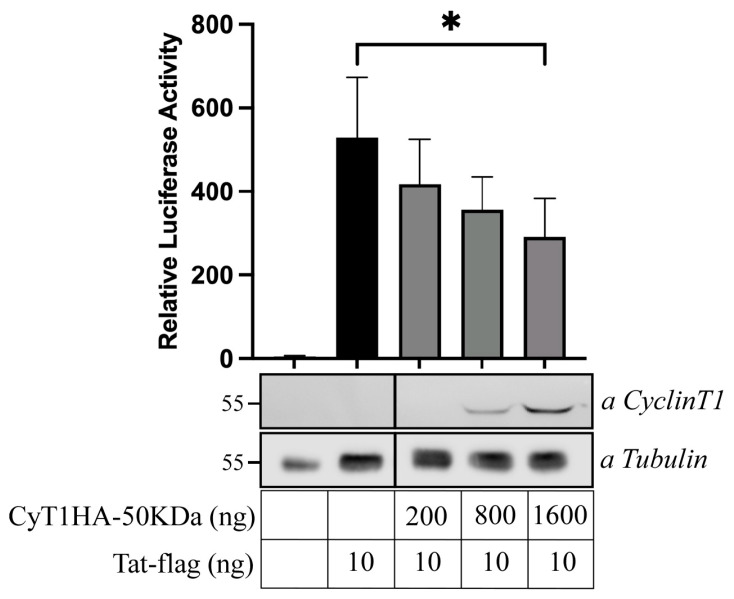
The 50 kDa Cyclin T1 inhibits the Tat transactivating function in U937 Minus. Increasing amounts (200, 400, and 1600 ng) of the expression vectors for 50 kDa Cyclin T1 were cotransfected in 293T cells with fixed amounts of plasmids pHIV-1 LTR-Luc, phRL-CMV, and pTat (10 ng). Mean luciferase activity, normalized to the *Renilla* activity, is presented. The activity of Tat is indicated by the black column. Column 1 represents the activity of the pcDNA3 vector. Results are the average of three independent experiments (* *p* < 0.05, unpaired *t* test). Error bars represent the standard deviation. The expression of for Cyclin T1 was assessed by Western blot by using anti-Cyclin T1 antibody. Tubulin was analyzed as a loading control.

**Table 1 viruses-16-01176-t001:** Sequence of the primers used to amplify CCNT1 gene.

Primer ID	Sequence
T1-FW	cac cgc cac cat gga ggg agg agg aag aac aac aac
T1-9RW	gct aaa ttc tca cta gtc cga tga ccc
T1-9FW	ggg tca tcg gac tag tga gaa ttt agc
T1-RW	ccc ggg cct cga gct ctt agg aag ggg tgg aag tgg tgg
T1-6FW	caa gca agg act tag cac aga c

## Data Availability

The data presented in this study are available on Appendix A.

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
