# Peer review of "A Truncated Isoform of Cyclin T1 Could Contribute to the Non-Permissive HIV-1 Phenotype of U937 Promonocytic Cells"

_viruses, 2024, doi:10.3390/v16081176_

Round 1

Reviewer 1 Report

Comments and Suggestions for Authors

This work investigates the difference in two isogenic clones of U937 cells that differ in their permissiveness to HIV infection. Previously termed “plus” and “minus” cells, for permissive and otherwise (respectively), the authors demonstrate that minus cells contain truncated forms of Cyclin T1, an important cellular cofactor required for efficient transcription from the provirus. Notably, these shorter forms can still interact with CDK9. RT-PCR of the RNA isolated from both types of cells failed to demonstrate different splice isoforms, suggesting that the regulation may not happen at the RNA level, but rather at the protein level. Cellular localization and transcriptional activation capability of this CycT1 form was investigated.

Major points

1. It’s interesting that CycT1 might be cleaved - or that a truncated form somehow exists - and that this might correlate with the permissiveness of these cells to HIV. This finding, however, begs the questions: How does this occur mechanistically? What does this mean for HIV infection or replication? The answers to these questions are lacking in the paper.

2. The authors talk a lot about the 50 kD species, but there seems to be a few species of truncated CycT1 as per the western blots (e.g. Fig 4). The smallest species even reappears under MG132 treatment. Presumably these are also CycT1 fragments, as they are detected with the same antibody, but it would be useful to know what these forms are and which parts of the canonical protein they contain.

3. How can one be sure that the nonpermissive phenotype is due to the 50 kD band and not others? In fact, in Fig 2A the more prominent species is the lower MW one. And while the 50 kD one is said to retain the capacity to interact with CDK9, it is unclear from Fig 2C that it is indeed this particular one that interacts with CDK9; it could be all of them, which brings me back to the previous point that it would be good to know what those additional species are and which parts of the protein they represent.

4. Fig 6b: are the construct names inadvertently switched? The upper panels co-stain with the nuclei, whereas the lower panels seem both cytoplasmic and nuclear. One would expect the truncated 50 kD form to be more diffuse in both compartments based on the text. (L328-333)

5. L85-86 and fig 7 suggest a dominant negative effect, but this has not been demonstrated in the actual model, only in terms of reporter assays. As the 50 kD form can be overexpressed, why not introduce this into U937 plus cells and see what happens to their HIV permissiveness?

6. As a role for cathepsins (or MMPs) are suggested in the processing of CyclinT1, could the authors do a quick experiment with a few protease family-specific inhibitors to narrow down the candidates? Or at least have some piece of data to indicate that this is indeed dependent on a cellular protease?

Minor points

I find the terms plus/minus is confusing in the title. Even if it’s explained in the text, it’s not instantly clear what promonocytic Minus cells means. Maybe one could say non-permissive U937 cells. Also, should minus and plus really be capitalized?

Typos etc:

L85: tat-mediated

L338: HA-tagged

L340: 50 kD

L342: polyclonal

L416: cathepsin

L437: tat-mediated

Comments on the Quality of English Language

Just minor typos. I pointed out the ones I found.

Author Response

This work investigates the difference in two isogenic clones of U937 cells that differ in their permissiveness to HIV infection. Previously termed “plus” and “minus” cells, for permissive and otherwise (respectively), the authors demonstrate that minus cells contain truncated forms of Cyclin T1, an important cellular cofactor required for efficient transcription from the provirus. Notably, these shorter forms can still interact with CDK9. RT-PCR of the RNA isolated from both types of cells failed to demonstrate different splice isoforms, suggesting that the regulation may not happen at the RNA level, but rather at the protein level. Cellular localization and transcriptional activation capability of this CycT1 form was investigated. 

Major points

  1. It’s interesting that CycT1 might be cleaved - or that a truncated form somehow exists - and that this might correlate with the permissiveness of these cells to HIV. This finding, however, begs the questions: How does this occur mechanistically? What does this mean for HIV infection or replication? The answers to these questions are lacking in the paper. 

We thank the reviewer for the comment. In the present work, we hypothesize that the truncated form of Cyclin T1 is generated by proteolytic cleavage performed by a specific protease expressed only in Minus cells. Unlike the full-length protein, the 50kDa cyclinT1, while retaining the ability to interact with CDK9, shows a different subcellular distribution, being expressed not only in the nuclear compartment but also in the cytoplasmic compartment. We suggest that this different distribution reduces the amount of CyclinT1 available for the formation of PTEF-b, which is the complex necessary for the promotion Tat-mediated HIV-1 transcription. This could represent a limiting factor for viral replication, at least under basal conditions. In the presence of gamma interferon stimulation, the amount of truncated CyclinT1 significantly decreases compared to the full-length form. This should in itself represent an advantage for the virus in Minus cells, in which, however, the increased expression of the Interferon gamma-induced restriction factors CIITA and TRIM22 results in the sequestration of the full-length CyclinT1 in TRIM22 bodies containing CIITA and in the consequent inhibition of viral transcription mediated by Tat. We have added a graphical abstract to clarify the proposed model in our work.

  1. The authors talk a lot about the 50 kD species, but there seems to be a few species of truncated CycT1 as per the western blots (e.g. Fig 4). The smallest species even reappears under MG132 treatment. Presumably these are also CycT1 fragments, as they are detected with the same antibody, but it would be useful to know what these forms are and which parts of the canonical protein they contain. 

Following the Reviewer suggestion, we also analyzed by mass spectrometry, the lower molecular weight bands (46 kDa and 32 kDa) immunoprecipitated with the Cyclin T1-specific antibody in Minus cells. Interestingly, all bands were enriched in peptides corresponding to the N-terminal region of Cyclin T1 containing the binding site for CDK9. LC-MS/MS data are reported in the revised Supplementary Table 1 and in the text (lines 246-250). We decided to focus on the 50 kDa band first because it is the most abundant in Minus cells, and second because it is likely that the lower molecular weight bands result from a further processing of the 50 kDa form (all the peptides belonging to the lower molecular weight bands are present in the 50 kDa form). This will be the focus of future investigation.

  1. How can one be sure that the nonpermissive phenotype is due to the 50 kD band and not others? In fact, in Fig 2A the more prominent species is the lower MW one. And while the 50 kD one is said to retain the capacity to interact with CDK9, it is unclear from Fig 2C that it is indeed this particular one that interacts with CDK9; it could be all of them, which brings me back to the previous point that it would be good to know what those additional species are and which parts of the protein they represent.

As explained above, the lower molecular weight bands (46 kDa and 32 kDa) are enriched in peptides corresponding to the N-terminal part of Cyclin T1 containing the binding site for CDK9. From figure 2A it is true that the more prominent band seems to be the 46 kDa one, but LC-MS/MS results showed that the 46 kDa band is significantly “contaminated” by Actin, as opposed to the 50 kDa band (see the revised Supplementary Table 1). However, we cannot exclude the possibility that the 46 kDa form is also capable of interacting with CDK9, as the interaction domain with CDK9 is located within the first 254 amino acids. We hypothesize that these truncated forms of CyclinT1 collectively contribute to the non-permissive phenotype of Minus cells. In this work, we focused on the 50 kDa band that was most significantly enriched with Cyclin T1 peptides. As mentioned in response to Reviewer’s comment n.2, the functional importance of the additional lower molecular weight bands will be the matter of future studies

  1. Fig 6b: are the construct names inadvertently switched? The upper panels co-stain with the nuclei, whereas the lower panels seem both cytoplasmic and nuclear. One would expect the truncated 50 kD form to be more diffuse in both compartments based on the text. (L328-333). We apologize for the mistake. The figure has been corrected.
  2. L85-86 and fig 7 suggest a dominant negative effect, but this has not been demonstrated in the actual model, only in terms of reporter assays. As the 50 kD form can be overexpressed, why not introduce this into U937 plus cells and see what happens to their HIV permissiveness? 

We thank the reviewer for the valuable suggestion. In this work, we focused on identifying the truncated form of Cyclin T1 in U937 Minus cells. To explain its possible role in acquiring the non-permissive phenotype of these cells, we conducted luciferase experiments by overexpressing the 50 kDa cyclin in 293T cells. Certainly, evaluating the impact that overexpression of the truncated Cyclin T1 might have in terms of "protection" from infection in Plus cells is extremely interesting. We plan to explore this further with the help of collaborators to conduct HIV-1 infection experiments. In addition, the transient transfection of U937 Plus cells is extremally low efficient compared to the one in 293T cells (5% to 70%, respectively). The generation of stable clones expressing high levels of 50 kDa CyclinT1 will be explored in the future.

  1. As a role for cathepsins (or MMPs) are suggested in the processing of CyclinT1, could the authors do a quick experiment with a few protease family-specific inhibitors to narrow down the candidates? Or at least have some piece of data to indicate that this is indeed dependent on a cellular protease? 

We agree with the reviewer that would be extremely interesting to identify the protease responsible for the cleavage of Cyclin T1. A precise characterization of the possible proteases will be also the focus of future investigation.

Minor points

I find the terms plus/minus is confusing in the title. Even if it’s explained in the text, it’s not instantly clear what promonocytic Minus cells means. Maybe one could say non-permissive U937 cells. Also, should minus and plus really be capitalized?

We changed the title as suggested by the reviewer. The new title is:”A truncated isoform of Cyclin T1 could contribute to the non-permissive HIV-1 phenotype of U937 promonocytic cells.” However, to be consistent with our previously published papers we named Plus and Minus cells in capital letter (Forlani G, Turrini F, Ghezzi S, Tedeschi A, Poli G, Accolla RS, Tosi G. The MHC-II transactivator CIITA inhibits Tat function and HIV-1 replication in human myeloid cells. J Transl Med. 2016 Apr 18;14:94. doi: 10.1186/s12967-016-0853-5.)

Typos etc: all the typos have been checked and corrected.

L85: tat-mediated

L338: HA-tagged

L340: 50 kD

L342: polyclonal

L416: cathepsin

L437: tat-mediated

Reviewer 2 Report

Comments and Suggestions for Authors

In this manuscript, Alberio et al. investigated why the Minus U937 cell line is nonpermissive to HIV-1 infection. The manuscript is easy to follow and investigates a mechanism of HIV-1 replication restriction that was not previously deeply described. However, several issues should be addressed before considering the acceptance of this manuscript for publication.

Minor comments:

-            There are several typos and grammatical errors (lines 45, 85, 86, 105, 114, etc.). The authors should carefully revise the manuscript and correct them.

-            Lines 126-136 and table 1, it might be redundant to specify the primers sequences both in the text and in a table. The authors should either choose to specify the primers sequences in the text or in the table. I suggest leaving the sequences in the table and refer to the primers by their ID in the text.

Major comments:

-            For the statistical analysis, do the variables analyzed follow a normal distribution? If so, then to use a t-test should be fine. However, the authors should perform a test to analyze the data distribution. If the distribution is not normal, the test use to analyze the data should be a non-parametric test.

-            The authors only test U937 cell lines, Minus and Plus, to demonstrate the mechanism by which HIV-1 infection and replication is restricted in the Minus U937 cell line. However, some experiments might be added to understand how this mechanism may affect HIV latency, by performing some analysis in U1 cell line, a latently infected cell line derived from the parental U937 cell line. Furthermore, it might be also interesting to test their findings in J-lat cell lines, as they also use Jurkat cell line in their experiments.

-            Another interesting experiment might be to try to infect with HIV the Minus U937 cell line by transfecting the 80 kDa Cyclin T1, or also silencing or transfecting the 50kDa form in the Plus U937 cell line and see if HIV-1 infection is restricted.

-            It also might be interesting to determine how this mechanism might be involved in HIV latency and persistence in primary cells, by analyzing HIV latency primary cell models or cells from people living with HIV.

Author Response

 Reviewer 2

In this manuscript, Alberio et al. investigated why the Minus U937 cell line is nonpermissive to HIV-1 infection. The manuscript is easy to follow and investigates a mechanism of HIV-1 replication restriction that was not previously deeply described. However, several issues should be addressed before considering the acceptance of this manuscript for publication.

Minor comments:

-            There are several typos and grammatical errors (lines 45, 85, 86, 105, 114, etc.). The authors carefully revised the manuscript and correct them. We read carefully the text and correct all the mistakes.

-            Lines 126-136 and table 1, it might be redundant to specify the primers sequences both in the text and in a table. The authors should either choose to specify the primers sequences in the text or in the table. I suggest leaving the sequences in the table and refer to the primers by their ID in the text. We thank the reviewer for the suggestion. We removed the primers’ sequence form the text and leave the information in Table 1.

Major comments:

-            For the statistical analysis, do the variables analyzed follow a normal distribution? If so, then to use a t-test should be fine. However, the authors should perform a test to analyze the data distribution. If the distribution is not normal, the test use to analyze the data should be a non-parametric test. The variables analyzed follow a normal distribution based on Shapiro-Wilk Normality test performed in Graphpad Prism. (p>0.05) In this case, t-test is applicable.

-            The authors only test U937 cell lines, Minus and Plus, to demonstrate the mechanism by which HIV-1 infection and replication is restricted in the Minus U937 cell line. However, some experiments might be added to understand how this mechanism may affect HIV latency, by performing some analysis in U1 cell line, a latently infected cell line derived from the parental U937 cell line. Furthermore, it might be also interesting to test their findings in J-lat cell lines, as they also use Jurkat cell line in their experiments.

In this work, we focused on the isogenic promonocytic Minus and Plus cell models, which differ in their susceptibility to viral infection. It is certainly interesting to investigate whether the findings presented here have a possible impact on HIV latency. Hower, we consider the above aspect, although important, out of the scope of the present investigation.

-            Another interesting experiment might be to try to infect with HIV the Minus U937 cell line by transfecting the 80 kDa Cyclin T1, or also silencing or transfecting the 50kDa form in the Plus U937 cell line and see if HIV-1 infection is restricted.

We thank the reviewer for the suggestion. However, our concern is that silencing the 50 kDa form of Cyclin T1 in Minus cells might not be optimal, as the active protease(s) would continue to produce the truncated form the full-length CiclinT1. A focus of our future investigation will be to identify the protease(s) in order to mask the cleavage site, thereby reducing the expression of the truncated form in Minus cells.

-            It also might be interesting to determine how this mechanism might be involved in HIV latency and persistence in primary cells, by analyzing HIV latency primary cell models or cells from people living with HIV.

The Reviewer suggestion is very interesting and will definitely be the next step in our research. The differential expression of Cyclin T1 in different patients could represent a marker of disease progression, and in the event of isolating the putative protease(s) responsible for the proteolytic cleavage, it could represent a pharmacological target to delay disease progression in HIV-positive individuals.

Reviewer 3 Report

Comments and Suggestions for Authors

The authors discovered a 50 kDa protein in U937 Minus cells (Figure 1), then isolated the protein by using a Co-IP assay, and identified that the protein is a truncated cyclin T1 by a mass spectrometry analysis (Figure 2). Then they showed that the 50 kDa Cyclin T1 protein is localized both the nucleus and cytoplasm and inhibits Tat transactivation (Figure 6 and 7).

Major comments:

1.     The paper used anti-CCTN1 antibody and CCNT1 gene for Immunoprecipitation and RT-PCR assays, please provide the full name of CCNT1, a brief introduction about CCNT1 and the reason of using CCNT1 for the assays.

2.     Most of the Figure results are provided with original images. However, the original image of α-tubulin of Figure 4 is not found. Please provide the original image of α-tubulin of the Figure 4 blots.  

3.     Is there a loading control for Figure 7 western blot?

Minor comments:

1.       Line 15: Please provide full name for all the abbreviations: TRIM22 and CIITA.  

2.       Line 34: Please provide full name of Tat

3.       Line 39: please provide abbreviation of RNA polymerase II, it used in line 329 as “RNA Pol II”

4.       Line 45: Please check PKC full name is protein kinase C or ” protein chinase C (PKC)”

5.       Please keep consistence of HIV-1 throughout the paper: Line 51, 57, 59: HIV-1

6.       Please keep a space between a number and a unit: i.e. line 83 “50kDa”

7.       Line 103: Please use the symbol “×” instead of “x” in line 103 “5x106 Minus and Plus cells” and wherever is relevant.

8.       Line 105: ‘with 0,1 % protease inhibitor…’→ with 0.1 % protease inhibitor…

9.       Please use ‘min’ and ‘h’ for the Mass spectrometry and Treatments sections (line 141- 167), please check throughout the paper.  

10.    Line 174: bovine serum albumin (BSA), BSA used in line 176

11.    Line 221: Figure 1B

12.    Line 320: Figure 6A ?

13.    Please keep reference format consistent. Reference 1-9, the journal names are in full, but most of the rest reference journal names are with their abbreviation formats.

Comments on the Quality of English Language

Please refer to above section.

Author Response

Reviewer 3

The authors discovered a 50 kDa protein in U937 Minus cells (Figure 1), then isolated the protein by using a Co-IP assay, and identified that the protein is a truncated cyclin T1 by a mass spectrometry analysis (Figure 2). Then they showed that the 50 kDa Cyclin T1 protein is localized both the nucleus and cytoplasm and inhibits Tat transactivation (Figure 6 and 7).

Major comments:

  1. The paper used anti-CCTN1 antibody and CCNT1 gene for Immunoprecipitation and RT-PCR assays, please provide the full name of CCNT1, a brief introduction about CCNT1 and the reason of using CCNT1 for the assay

We added these information in the material and methods section, line 125.

  1. Most of the Figure results are provided with original images. However, the original image of α-tubulin of Figure 4 is not found. Please provide the original image of α-tubulin of the Figure 4 blots.

We reuploaded the original image, in which a-tubulin blot is present.

  1. Is there a loading control for Figure 7 western blot?

 We put the immunoblot for a-tubulin as loading control of the experiment

Minor comments: all the typos have been checked and corrected.

  1. Line 15: Please provide full name for all the abbreviations: TRIM22 and CIITA.  
  2. Line 34: Please provide full name of Tat
  3. Line 39: please provide abbreviation of RNA polymerase II, it used in line 329 as “RNA Pol II”
  4. Line 45: Please check PKC full name is protein kinase C or ” protein chinase C (PKC)”
  5. Please keep consistence of HIV-1 throughout the paper: Line 51, 57, 59: HIV-1 
  6. Please keep a space between a number and a unit: i.e. line 83 “50kDa”
  7. Line 103: Please use the symbol “×” instead of “x” in line 103 “5x106 Minus and Plus cells” and wherever is relevant. 
  8. Line 105: ‘with 0,1 % protease inhibitor…’→ with 0.1 % protease inhibitor…
  9. Please use ‘min’ and ‘h’ for the Mass spectrometry and Treatmentssections (line 141- 167), please check throughout the paper.  
  10. Line 174: bovine serum albumin (BSA), BSA used in line 176
  11. Line 221: Figure 1B
  12. Line 320: Figure 6A ?
  13. Please keep reference format consistent. Reference 1-9, the journal names are in full, but most of the rest reference journal names are with their abbreviation formats.

Round 2

Reviewer 2 Report

Comments and Suggestions for Authors

In this manuscript, Alberio et al. investigated why the Minus U937 cell line is nonpermissive to HIV-1 infection. The manuscript is easy to follow and investigates a mechanism of HIV-1 replication restriction that was not previously deeply described. After the first revision, the authors have addressed all the reviewers' comments, and the manuscript might be accepted for publication.